# Internet Traffic Classification with Federated Learning

**Hyunsu Mun and Youngseok Lee \***

Department of Computer Engineering, Chungnam National University, 99 Daehak-ro, Yuseong-gu,
Daejeon 34134, Korea; munhyunsu@cnu.ac.kr
**\*** Correspondence: lee@cnu.ac.kr

**Abstract:** As Internet traffic classification is a typical problem for ISPs or mobile carriers, there have been a lot of studies based on statistical packet header information, deep packet inspection, or machine learning. Due to recent advances in end-to-end encryption and dynamic port policies, machine or deep learning has been an essential key to improve the accuracy of packet classification. In addition, ISPs or mobile carriers should carefully deal with the privacy issue while collecting user packets for accounting or security. The recent development of distributed machine learning, called federated learning, collaboratively carries out machine learning jobs on the clients without uploading data to a central server. Although federated learning provides an on-device learning framework towards user privacy protection, its feasibility and performance of Internet traffic classification have not been fully examined. In this paper, we propose a federated-learning traffic classification protocol (FLIC), which can achieve an accuracy comparable to centralized deep learning for Internet application identification without privacy leakage. FLIC can classify new applications on-the-fly when a participant joins in learning with a new application, which has not been done in previous works. By implementing the prototype of FLIC clients and a server with `TensorFlow`, the clients gather packets, perform the on-device training job and exchange the training results with the FLIC server. In addition, we demonstrate that federated learning-based packet classification achieves an accuracy of 88% under non-independent and identically distributed (non-IID) traffic across clients. When a new application that can be classified dynamically as a client participates in learning was added, an accuracy of 92% was achieved.

**Keywords:** federated learning; traffic classification; privacy

## 1. Introduction

Internet traffic classification is a representative research topic that has been significantly studied. There are three typical approaches: the first is to examine packet header information such as IP address and port numbers; the second is to look into the packet payload to find the bit signature; the third is to utilize the machine learning method. Packet header information is useful for firewalls and intrusion detection systems to find the traffic pattern. However, the packet payload inspection is confronted with a challenge as web and mobile applications usually exchange encrypted packets through TLS/SSL for user privacy. Insteads, machine learning technologies have been widely applied to Internet traffic classification research.

Machine learning, which is used in various fields such as Assistive Robotics, recommender system, design and deployment of UAV, has been studied to be effective in classifying Internet traffic [1–4]. While machine learning is useful for Internet traffic classification under packet encryption, ISPs or mobile carriers need to collect application packets and their information from users for the training process. Because users' packet data and application usage information are private, it is impossible to collect user traffic at the central server. Thus, machine learning-based traffic classification still faces the prerequisite challenges of the privacy issue in traffic collection.

The recent development of machine learning evolves towards the distributed environment. One of the promising solutions is so-called federated learning that trains user data on the device and exchanges only training models and their update information [5–7]. In federated learning, the central server computes only the aggregated average of the training data gathered from each device. Therefore, it protects privacy as the user data resides on the device.

Although federated learning is promising because of privacy concerns, it has not been fully understood for the feasibility of Internet traffic classification under the realistic data model with unbalanced, independent and individual distribution (IID) characteristics. For instance, employing federated learning to traffic classification requires solving following challenges.

- Federated learning traffic classification framework: Because learning works on the client and the server gathers the clients' training results and aggregates the average, a federated learning traffic classification framework on distributed training, traffic collection, and training result aggregation are needed.
- Federated learning communication protocol: As diverse clients are irregularly connected to a server and they are often disconnected, a fault-tolerant client-server communication protocol that deals with connection management and efficient client selection should be devised.
- Federated learning traffic classification algorithm under non-IID and unbalanced data: In federated learning, the non-IID and unbalanced data for traffic classification have to be considered. Because each client uses diverse applications, their traffic and application information are different from each other.

In this paper, by considering the above requirements, we propose a federated-learning Internet traffic classification framework (FLIC) that can label packets into applications dynamically. In FLIC, the federated client directly trains its own packet data on the device, and sends only the result to the federated learning server. Hence, FLIC can protect the user's privacy regarding packets. Besides, FLIC supports the dynamic federated learning model that can learn the labels of new applications, which has the strength compared with the previous work assuming the static Internet application categories.

We run an experiment on Internet traffic application classification using the IS-CXVPN2016 traffic data generated in [8] to examine the characteristics of Internet traffic [9]. From the experiments, we demonstrate that the accuracy of FLIC is comparable to the centralized learning method, and its accuracy increases in proportion to the number of clients. Also it is shown that when a federated learning participant with new applications joins, FLIC can quickly update the global learning model for every federated clients to classify new applications. In the environment of non-IID traffic distribution and of dynamically increasing clients, FLIC achieved 88% and 92% accuracy, respectively.

## 2. Related Work

Recently, a method using machine learning models has emerged for classification of internet traffic. Internet traffic classification that can be used in various fields such as firewall and monitoring has been studied by methods such as HTTP header monitoring and deep packet inspection (DPI). However, as SSL/TLS-based encrypted communication became popular, it became difficult to use existing Internet traffic classification techniques. In this situation, not only classifying internet traffic by IP-TCP/UDP layer or man-in-the-middle (MITM), but also a method using a machine learning model attracted attention. Lotfollahi et al. [10] proposed a method of using autoencoder for Internet traffic classification. Although it is a CNN that works well for classifying structured data, the problem that it is difficult to know which features to make into images was solved by using autoencoders. However, since the average accuracy of autoencoder is slightly lower than that of CNN, there remains the possibility of improvement such as combining the two. Wang et al. [11] proposed dedicated classifiers using 1D-CNN and autoencoder after comparing the accuracy of autoencoder, CNN, and RNN when classifying internet traffic using CNN.

Wang et al. [12] compared the performance of 1D-CNN and 2D-CNN in Internet traffic classification, and 1D-CNN showed higher accuracy by extracting features better. In this paper, the problem of allowing the CNN to easily understand the structure of the packet is to vectorize the packet by using the TCP/UDP-IP packet structure without using an autoencoder.

Internet traffic classification based on deep learning uses a convolution neural network (CNN) or a recurrent neural network (RNN) method. Though RNN that considers the temporal order of feature vectors is employed to classify malicious traffic in [13], it is widely reported that the CNN model can more clearly extract structural features of Internet traffic [11]. Therefore, we adopt the CNN model for relatively better accuracy in packet-level classification of Internet traffic applications.

Recently, federated learning, a method in which a client trains a model with its own data locally, and delivers only the training result to the federated server, has been proposed [14]. Research using this method is being actively conducted because it can solve the problem of using edge client computing power and protecting sensitive data privacy. In particular, since Internet traffic data is sensitive data that contains the user's personal information, a technology that can learn a model while protecting the client's personal information is required. Google has announced a keyboard recommendation application using federated learning, called G-board [7]. it predicts and suggests the next word to be used by the user through mobile keyboard typing data, showed that sensitive data called user keyboard typing data can be developed by protecting personal information through federated learning.

Since federated learning or federated optimization was proposed as a way to learn a global model using data distributed over an extremely large number of nodes [15], its performance is related to client selection and training data distribution. Konečný et al. [15], who proposed A distributed version of SVRG, said that it works well in a simple distributed learning environment, but that there are still problems to be solved in situations where data is sparse. After that, McMahan et al. [16] proposed a federated average algorithm and showed that the number of local minibatch and local epoch in federated client affects the learning speed. In particular, federated average (FedAvg) works even in non-IID data environments.

In federated learning, client selection is a very important factor that affects not only behavior but also accuracy. Bonawitz et al. and Nishio et al. [17,18] proposed a system and an algorithm for selecting a client to participate in learning, and showed how to consider the situation such as power supply and computing power of distributed clients. Nishio et al. [18] secures availability by setting a deadline time for training completion and selecting a client that can complete training in it. Since this method assumes a cellular network that can be easily disconnected, an environment where clients can quickly participate in learning was considered. However, in the prediction of the learning time in the client, there are many things to consider, such as mobile computing power, the amount of data that the client has, and the amount of power in the client, so there remains a problem to be solved.

In addition to the client selection, the distribution of data also has a significant impact on federated learning. Client devices participating in federated learning use their own data for learning without transmitting it to a central server, and the stored data is characterized by non-IID (non independent and identically distributed random variables). Zhao et al. [19] shows that the federated learning accuracy is decreased by splitting the MNIST data into a non-IID form, proposing a solution through data sharing. The data sharing technology for improving accuracy in a non-IID environment proposed by [19] can be operated by a client transmitting data to a federated server as a kind of donation concept. Because accuracy can be secured by reducing the weight diversity between clients, it can be used in less sensitive data environments. when [20] determines the learning weight dynamically, ensuring the accuracy of federated learning in sparse data and poorly balanced classes, such as mobile keyboard data.

As federated learning methods considering client selection and non-IID environments are actively studied, research to classify Internet traffic as federated learning has also been conducted. Bakopoulou et al. [21] proposed a technology to identify the presence of personal information and advertisements by extracting HTTP text data through MITM in a mobile environment. Since MITM is performed using Local VPN, user's personal information is processed in a decrypted state only inside the device. However, HTTP Public Key Pinning (HPKP) technology is used, and there is a situation where the application does not work when decryption using VPN is performed. To solve this problem, a method of classifying Internet traffic without decryption is required.

On the other hand, Zhao et al. [22] proposed a method of using a common layer for each task when using federated learning for network anomaly detection, VPN or Tor traffic recognition, and traffic classification. This method guarantees higher accuracy by sharing the layer that extracts features and allowing learning with more data.

## 3. Federated Learning-Based Internet Traffic Classification (FLIC)

In this section, we explain the overview of federated learning-based Internet traffic classification (FLIC) architecture.

### 3.1. Architecture

Figure 1 describes the overview of the FLIC protocol. FLIC eliminates the need to collect packets from clients in order to classify Internet traffic using deep learning such as convolution neural network (CNN). In federated learning, the federated server is responsible for managing the global model and responding to requests from participating clients for training or model use. As the FLIC server does not gather client packet data, FLIC protects personal information contained in packets, and it reduces the overhead of a centralized server for data processing and communication. The federated client collects data and trains the model downloaded from the federated server on the device.

Internet traffic classification with FLIC is divided into four steps between the federated server and clients. As shown in Figure 1, the federated client downloads the global model and its parameters for the initialization (1). Then, the FLIC client trains the global model locally with its own data (2), and the federated server receives only the training results from clients (3) and aggregates them (4). The FLIC client classifies packets into the application labels received from the server (5).

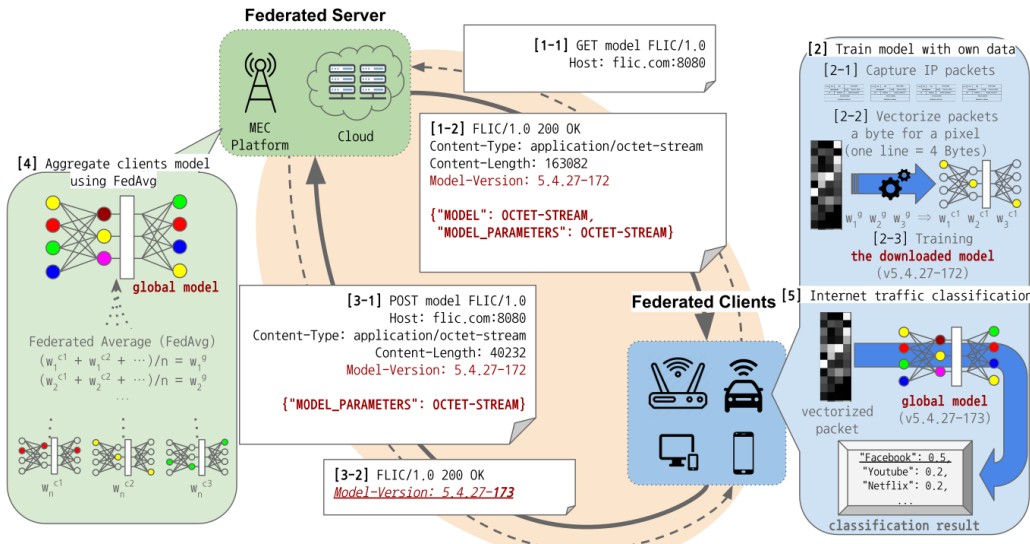

**Figure 1.** The architecture of federated learning-based Internet traffic classification (FLIC).

Federated clients must initialize training parameters and download the model to perform Internet traffic data collection, model training, and application classification

(`MODEL, MODEL_PARAMETERS`). Since the global model requested by the client includes application information used for training as well as global weights, parameter initialization and model downloading are essential for the client. The global model version contained in the response header (`Model-Version: 5.4.27-172`) is the key element when a federated client participates in training, and it is explained in detail in Section 3.2.

After downloading the global model and initializing the parameters, the federated client captures IP packets for Internet applications, vectorizes packets, and performs model training in order to participate in the training. A client can know the port number used by applications included in the global model through tools such as `netstat` and `ss`. Thereafter, the client collects IP packets for each application with `tcpdump` or `tshark`, and it stores packets in a `pcap` file. Before learning the collected data, the client pre-processes the packet vector of a $4 \times 375$ pixel image. A packet vector image is not transmitted to the central server, and it is used for training the model received from the federated server.

The client sends only the parameters of the model trained with its own data to the federated server. The header of the upload request specifies the model version for training, and it prevents the federated server from merging weights from the previous versions. After receiving the model weights, the server updates the revision number (`Model-Version: 5.4.27-173`) in the model version and delivers it to the clients participating in the training. The client with this revision number does not attend the training with the same model version, because it carried out the training job before.

When receiving the model weights from the federated clients, the federated server merges them into global model weights using the federated average algorithm. Unlike previous federated learning with the stateful connection management, which performs client selection, FLIC operates in the stateless manner. As the FLIC server does not know how many clients will participate in learning, it cannot compute the average of the weights with the number of clients participating in the training ($\sum w_i$ and $n$). To solve this problem, we include a revision number that matches the number of clients participating in the training in the global model version (`Mode-Version: 5.4.27-172`). Therefore, even if a client asynchronously uploads a new training result, the federated server can run the federated average algorithm because the number of clients participating in the training can be checked with the revision number of the global model.

Through the global model version including the revision number, federated clients are able to know the latest model for classifying Internet traffic applications. Each client downloads the latest global model from the federated server to perform Internet traffic application classification. The client captures IP packets with `tcpdump` or `tshark`, and it vectorizes them for packet-level application classification. The IP packet classification result can be used for configuring the firewall policy or recording user digital activities.

The stateless federated learning protocol through the model version used by FLIC responds well to exceptions such as client disconnection and learning failure, but has a weakness of slowing the overall learning speed. In particular, if the round progresses while the client is learning, all the cost of the corresponding learning operation is discarded, leading to a waste of computing power. The weakness that comes from this stateless protocol is one of the problems to be solved later.

### 3.2. Protocol

Table 1 explains the FLIC protocol message based on HTTP. The `GET` or `POST` methods are used to deliver the model data, `MODEL_BINARY` and application labels, `LABEL_JSON`. To eliminate the overhead of federated learning server for client connection management, we design the FLIC protocol based on HTTP RestFul API in the stateless manner. The FLIC protocol maintains the model version information with which each client decides whether to participate in the current learning stage or not.

The model version in FLIC consists of Global (G), Applications (A), Rounds (R), and revision (r), which is expressed in `G.A.R-r`. In federated learning, if a small number of clients participate in training, the reduced amount of collected data will result in the

poor accuracy. Therefore, the server should efficiently control clients participating in the learning. We minimize the overhead of managing the connection between the server and clients by using the simple model version format in the stateless manner. The G, A, and R version values are updated when the server changes the model structure, adds a new application, or proceeds to the next training round to improve the accuracy. Each client remembers the G, A, and R version values of the model with which it participates in training when the model version is updated. The server issues a revision values to the client participating in the training, and the client examines that the model version it has trained for the local model is the latest version.

**Table 1.** FLIC protocol description.

| Fields | | Description | Example |
|---|---|---|---|
| REQUEST | Method | Indicate the desired action | GET, POST |
| | Resource | Identify the target resource | model, labels |
| RESPONSE | Status Code | Summarize the response status | 200, 404 |
| | Textual Reason | Explain the reason for the answer | OK, Not Found |
| HEADER | Host | The domain name of the server and port number | example.com:8080 |
| | Content-Type | The media type of the body of the request | json, octet-stream |
| | Content-Length | The length of the request body | 158 |
| | Model-Version | The model version (G.A.R-r) | 5.4.27-172 |
| | Global (G) | The version of model structure | 5 |
| | Applications (A) | The version of application labels | 4 |
| | Rounds (R) | The rounds of federated learning on current model | 27 |
| | Revision (r) | The number of participated clients for this rounds | 172 |
| BODY | Model data | The binary model data | MODEL_BINARY |
| | Application labels | Represents a list of trainable application labels | LABEL_JSON |

Federated clients and the server exchange the FLIC message including the current model version as shown in Figure 2. When a FLIC client joins in the training, it first downloads the current model version, model data and application labels from the server with the GET message (Figure 2a). A model version includes the model layer, the application labels, training rounds attributes. After training the model with its own data, the FLIC client will send the model to the server with the POST message. Then, the FLIC server aggregates the model with Federated averaging (FedAvg). FedAvg is a generalization of FedSGD, which allows local nodes to perform more than one batch update on local data and exchanges the updated weights rather than the gradients.

When a FLIC client collects new application data not in the current application labels received from the FLIC server, it sends the application label information using POST method to the server (Figure 2b). When the server receives the new application data information, it dynamically updates the classification label information of the global model, while maintaining the parameters of the convolution layer for feature extraction and changing only the dense layer for classification. Then, other federated clients learn the global model parameters including new labels that have not been observed before on its device. At every federated learning round, all federated clients update the new application information.

### 3.3. Training Model and Feature Vector

Since a packet is a byte stream, FLIC converts a group of packets into feature vectors for deep learning. As CNN is useful for expressing the structural learning feature for a group of packets, FLIC employs CNN for packet classification with the specific header structure implying the corresponding communication protocol. In order to build the visible image of the communication protocol structure, we transform a packet into the image data of 4 pixels in width. This packet-to-vector image conversion method explains the packet structure in 4-byte blocks, which is useful for learning in the CNN model.

Figure 3 is the packet-to-vector image of ISCXVPN2016 data, which is widely used in Internet traffic classification studies. Figure 3 shows only the first 100 bytes of samples . Since the TCP/IP header of a packet is converted to a feature vector, the top 10 pixels are usually composed of 20 bytes in the IP header (top 5 pixels) and 20 bytes in the TCP header (next top 5 pixels), which represent the characteristics of the packet header. A vector of 365 pixels is built in the payload if there is no option in the packet header.

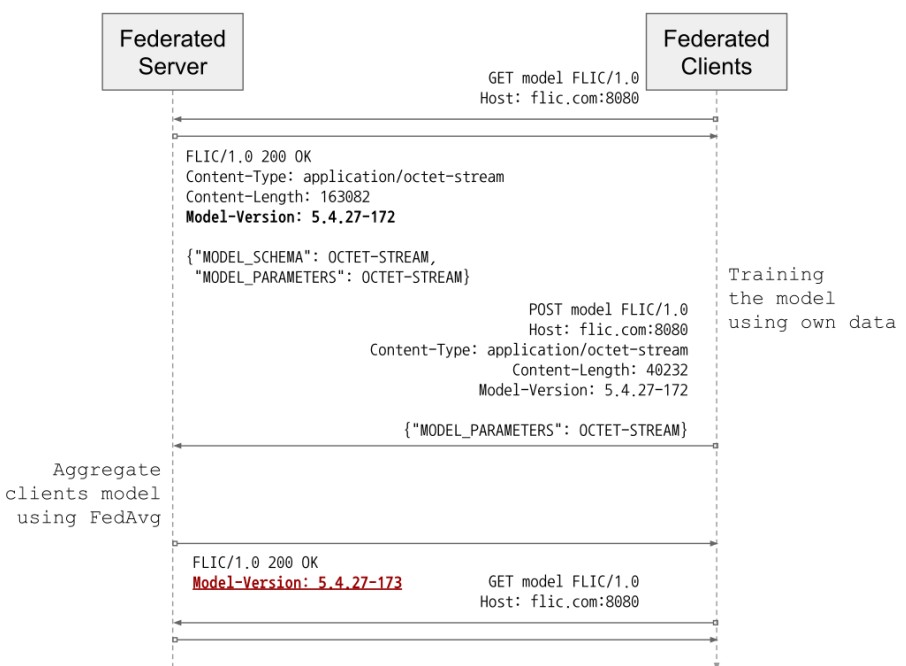

(**a**) Client initialization by downloading the model and joining the training process

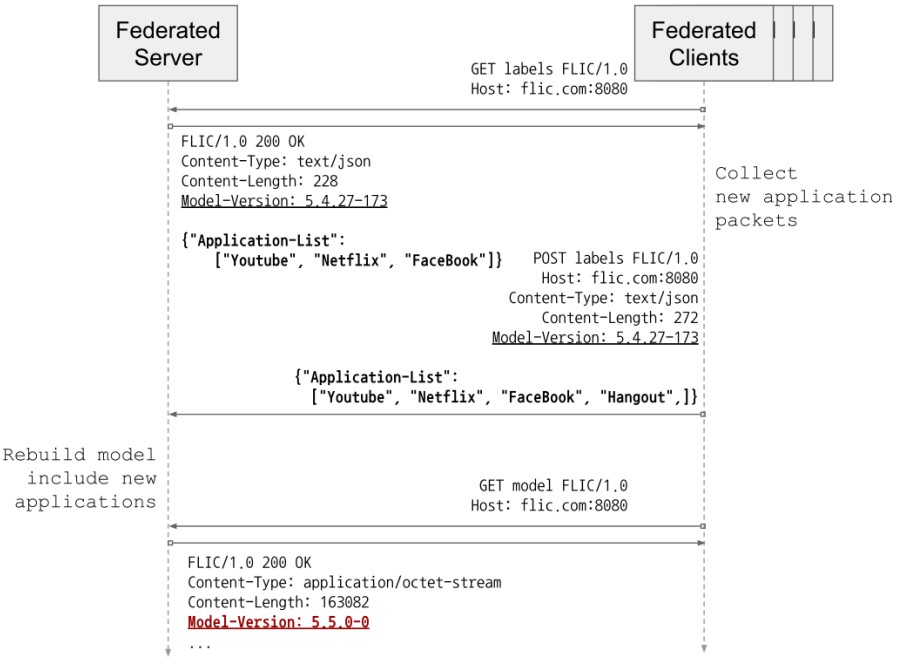

(**b**) Rebuilding the training model with new application labels

**Figure 2.** A federated-learning Internet traffic classification protocol (FLIC): initializing, updating, and rebuilding the training model.

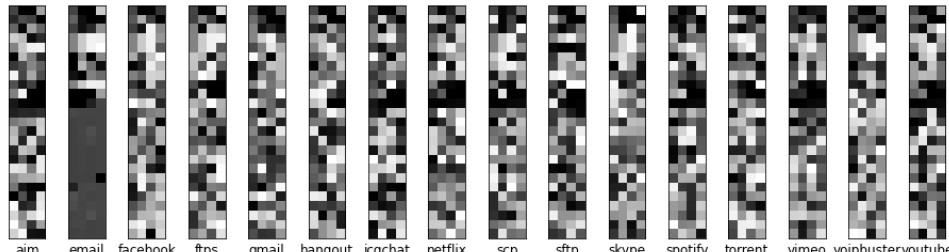

**Figure 3.** Packet-to-vector images used for packet classification: 16 application images in 100 bytes with payload (ISCXVPN2016 dataset).

The CNN model is used because it can effectively learn the structural characteristics of packets for Internet traffic classification as shown in Figure 4.

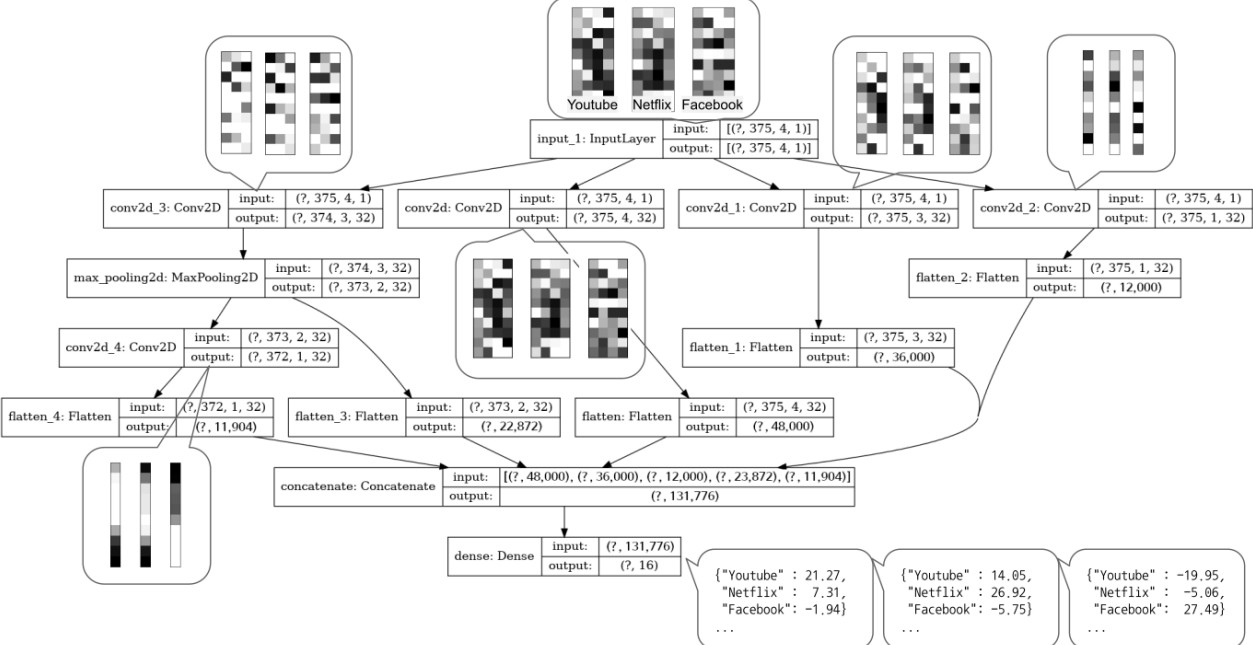

**Figure 4.** Convolution neural network model for classification of Internet traffic applications.

In particular, the model focuses on the TCP/IP header protocol including IP header address and TCP/UDP port numbers, which reflect the main features of the packet. It utilizes the first 1, 2, and 4 bytes of a packet, respectively for the convolution layers of sizes (1, 1), (1, 2), and (1, 4). In order to learn and identify the packet payload pattern, a (2, 2) size convolution layer is also added.. This deep and wide model clearly detects TCP/IP packet headers and payload patterns in units of 1, 2, and 4 bytes.

Additionally, the model consider a (2, 2) convolution layer and a pooling layer to find features in the payload. In particular, when a packet containing a file magic number of specific data structure, its vector will show the visual pattern. This feature is extracted and transformed into a 2D convolution layer and a pooling layer. As a result, our model highlights the features of the IP packet header with the 1D convolution layer and the features of the payload with the 2D convolution layer.

### 3.4. Federated Optimization

Federated learning performs training on each client device. As federated learning relies on stochastic gradient descent (SGD) for optimization, the objective of FLIC is given in Equation (1). Federated averaging (FedAvg) is a generalization of FedSGD, which allows local nodes to execute more than one batch update on local data and exchanges the updated weights rather than the gradients [15,16].

$$\min_{w \in \mathbb{R}^d} f(w) \quad where \quad f(w) \overset{def}{=} \frac{1}{n} \sum_{i=1}^{n} f_i(w) \tag{1}$$

where $f_i(w)$ is the loss of the prediction, $l(x_i, y_i : w)$, for example $(x_i, y_i)$ made with the model parameter $w$. Assuming $K$ clients over which the data is partitioned $P_k(n_k = |P_k|)$, the objective (1) is rewritten as follows.

$$f(w) = \sum_{i=1}^{K} \frac{n_k}{n} F_k(w) \quad where \quad F_k(w) = \frac{1}{n_k} \sum_{i \in P_k} f_i(w) \tag{2}$$

When assuming IID data, training over each FLIC client's data is equally contributed to the global model with $\frac{1}{n_k}$. In general, the relative influence ratio of each FLIC client according to unbalanced and non-IID data can be defined by considering the sample space for each client ($\frac{n_k}{n}$).

However, in FLIC based on stateless connection management, $n_k$ and $n$ asynchronously increase whenever the client uploads the learning result. Therefore, the FLIC server stores a revision number indicating the number of clients participating in the training in the global model version. Due to the revision number, even if a learning result is uploaded asynchronously, $n$ can be easily inferred and the federated average can be easily calculated. The client delivers $n_k$ representing the size of the dataset in the JSON body of the POST message. The (revision, $n_k$) mapping table of each client is be stored in the federated server.

### 3.5. Dynamic Classification

FLIC has the dynamic Internet traffic classification feature that adds new application labels during the training process, whereas traditional Internet traffic classification studies support only the static categories. Due to dynamic classification, all FLIC participants can learn new application categories on-the-fly.

With IP packets, the feature extraction layer can be recycled because packet headers standardized in 1, 2, and 4 bytes are usually arranged in the same order. Due to the TCP/IP header byte order feature, we can add a new Internet application label to the already learned model. As shown in Figure 5, when a client reports that new application data has been observed, the federated server updates the global model so that the new application can be classified. Therefore, FLIC dynamically adds new application targets.

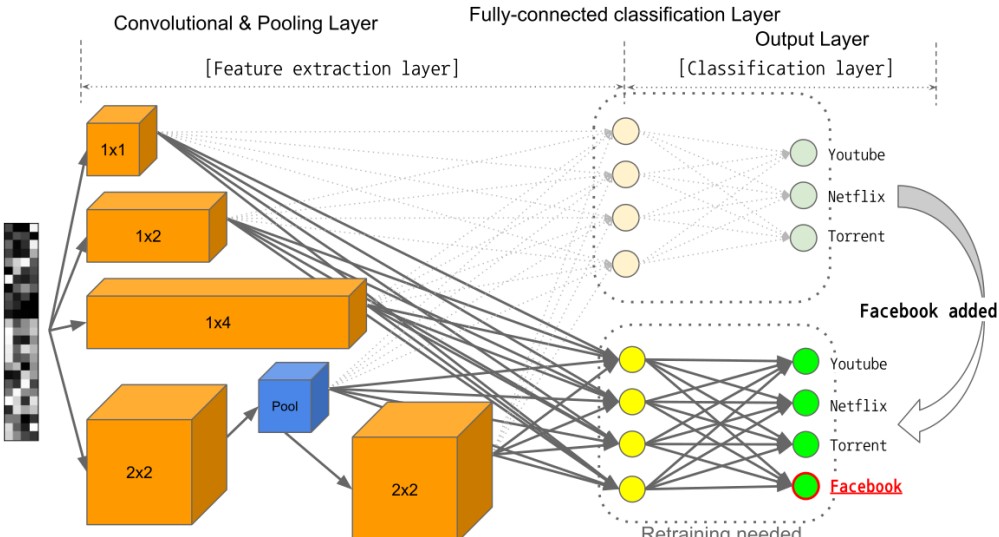

**Figure 5.** Model transformation when a new application is added (Facebook): FLIC keeps the weight of the feature extraction layer, but trains the classification layer weight for the new application.

When a new application is observed, the feature extraction layer is fixed by utilizing the characteristics of the structured packet. Then, learning only the classification layer, FLIC secures the accuracy with fewer federated communication rounds. The dynamic application classification capability of FLIC is based on learning similar types of data in advance and fine-tuning. The experimental results are explained in Section 4.6.

## 4. Performance Evaluation

In this section, we examine the performance evaluation results of FLIC. In federated learning, since each client participates in learning with its own data, the overall learning accuracy varies according to the number of clients and data distribution. Under the stochastic gradient descent optimizer, we carry out federated learning in the iterative manner under the variable client sets and non-IID data.

The experiment runs through the data preprocessing module, global model generator, and learning simulator. We preprocess and store sampled data with `TFRecords`. The preprocessing module reads `pcap` data using scapy and stores it in image. If the image is saved in `jpg` or `png`, loss may occur. The image is saved in 2D-array format like `numpy.ndarray`. After preprocessing, the data is partitioned to be distributed to clients for each experiment into two IID datasets and two non-IID datasets.

We use `TensorFlow` for FLIC performance evaluation under the different number of clients, non-IID environment, and dynamic applications. The global model generator creates the model shown in Figure 4 and initializes the model parameters. The generator creates a model with not only the feature extraction layer but also the classification layer. Therefore, in the dynamic application experiment, the generator must initialize by copying the weights of feature extraction layer from the existing model.

We preprocess and store sampled data with `TFRecords`, and distribute samples to federated clients. We implement the simulator using the `TensorFlow Federated simulation API`, we test federated learning with partitioned data and global models. The simulator distributes the partitioned data to federated clients to each client, delivers the global model, and waits for a training command. When the learning process starts, each client learns only with its own data and then uploads the learned model through the simulator. The simulator, which receives the training model result from the client, updates the global model with the federated average algorithm. `Stochastic gradient descent (SGD)` function was used as the optimizer of ferderated client and server. Each learning rate was set to 0.02 and 1.0.

Experimental data generation and federated learning experiments are run in Jupyter notebook which is available on our Github project [23].

### 4.1. Datasets

The `ISCXVPN2016` packet dataset is used to evaluate the accuracy of FLIC classifying Internet applications (Table 2). We vectorize each packet with the payload of 16 applications into images. For the realistic experiment, we sample packets from each application with different weights reflecting the popularity and the usage. The popular applications such as `YouTube`, `Netflix`, and `Facebook` have high weight (34) and `SCP`, `FTP`, and `ICQChat` have lower weights (1).

We run experiments by organizing the sampled packets into four datasets. Then, the IID datasets are grouped into A and B, and the non-IID datasets are grouped into A and B too. Basically, The sampled data is divided into 10% for the test data set and remaining 90% is distributed to a set of clients (16 to 100).

Figure 6 shows the packets distribution of 16 applications. From 15,651,658 packets with payload, 163,328 packets are sampled for each application. By varying the weight of application, we prepare the realistic sample packets for each application. For example, `FTPS`, `SFTP`, and `SCP`, which have a large number of data packets but are not commonly used, have the decreased amount of sample packets. `ICQChat` has a small number of 638 packets with the payload, but it is included to train applications with the small amount of packets in FLIC.

**Table 2.** Datasets for experiments (16 applications).

| | AIM<br>SCP | Email<br>SFTP | Facebook<br>Skype | FTPS<br>Spotify | Gmail<br>Torrent | Hangout<br>Vimeo | ICQChat<br>VoIPBuster | Netflix<br>YouTube |
|---|---|---|---|---|---|---|---|---|
| Total Packets | 4766<br>825,510 | 38,173<br>864,931 | 1,874,975<br>3,237,937 | 7,872,821<br>40,862 | 12,239<br>108,498 | 3,158,535<br>146,296 | 4118<br>842,495 | 299,166<br>251,866 |
| Packets<br>with payloads | 4296<br>212,599 | 35,713<br>782,227 | 1,725,927<br>2,659,033 | 5,911,511<br>22,268 | 11,003<br>72,487 | 2,979,870<br>92,962 | 638<br>841,878 | 160,529<br>138,717 |
| Sampled<br>packets | 638<br>1276 | 6380<br>1276 | 14,036<br>10,208 | 1276<br>17,864 | 8932<br>21,692 | 10,208<br>11,484 | 638<br>14,036 | 21,692<br>21,692 |
| [Sections 4.2 and 4.5]<br>IID dataset A | Sampled packets per application are evenly distributed to 32 clients. | | | | | | | |
| [Section 4.3]<br>IID dataset B | Sampled packets per application are evenly distributed to 100 clients. | | | | | | | |
| [Section 4.4]<br>Non-IID dataset A | Sampled packets per application are evenly distributed to 1 to 8 clients (Total 16 clients).<br>Each client has a different number of applications from 1 to 8. | | | | | | | |
| [Section 4.6]<br>Non-IID dataset B | Sampled packets per application are evenly distributed to 2 clients (Total 16 clients).<br>Each client has two applications. | | | | | | | |

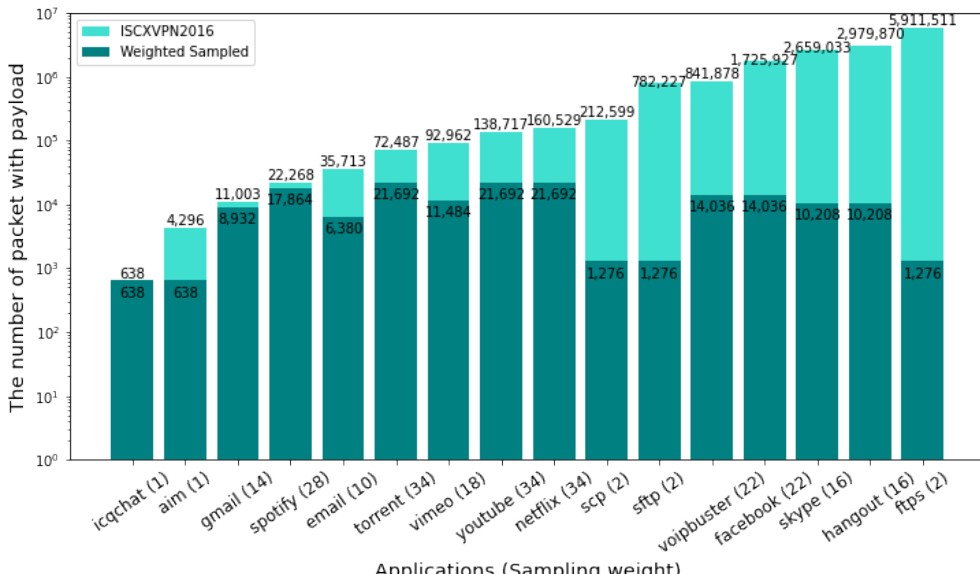

**Figure 6.** Samplesof test packets (ISCXVPN2016 dataset).

### 4.2. Centralized vs. Federated Learning

We run a comparative experiment to prove the possibility of Internet traffic classification with FLIC (IID dataset A in Table 2). The experiment compares the accuracy between federated learning-based Internet traffic classification and centralized learning. As shown in Figure 7, only 2% of the accuracy difference is observed between the centralized learning (epoch 100) and the federated learning of 32 clients (local epoch 1 and round 100). In particular, compared to centralized learning, federated learning has a smooth form in which the loss per round is reduced, which means that it will take longer time to complete learning. However, when learning is completed, the accuracy of both federated learning and centralized learning is higher than 95%, resulting in no significant difference.

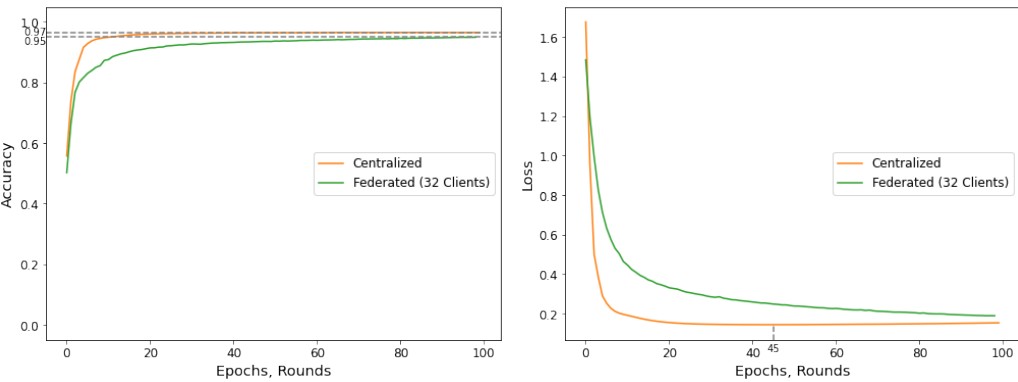

(**a**) Accuracy by epochs or rounds                                       (**b**) Loss by epochs or rounds

**Figure 7.** Comparison between centralized learning and federated learning for Internet traffic application classification (32 clients and 100 rounds for FLIC).

The orange line in Figure 7 is the accuracy and loss plot of centralized learning, and overfitting has occurred from epoch 45. The green line keeps increasing by epoch, which means federated learning achieves the improvement of accuracy by epoch. As securing 97% accuracy, the experiment do not deal with the overfitting by adjusting the learning rate. In addition, it set the learning variable of federated learning, the local epoch, which affects the learning completion speed, to 1 for comparison with centralized learning.

Figure 8 shows the accuracy of each application with the confusion matrix of centralized learning and federated learning. Compared to centralized learning (Figure 8a), federated learning progresses its learning slowly under a small amount of packet data (Figure 8b). This is because information about applications is not enough at first under the small data when creating a global model with the federated average algorithm. However, as the round progresses, the accuracy of the global model improves when the client repeats training based on the global model.

### 4.3. Accuracy by the Number of FLIC Participants

When classifying Internet traffic through federated learning, the accuracy is affected by the number of FLIC clients. In order to observe how the number of clients influences the accuracy of Internet traffic classification, we perform an experiment by changing the number of clients with the total amount of proportional data records (IID dataset B in Table 2). Figure 9 shows that as the number of clients increases up to 100, the accuracy of Internet traffic classification is enhanced.

In federated learning, as the number of clients increases, the amount of data available for learning grows. Under a large volume of data, the accuracy and learning speed of Internet traffic classification are improved. The data is divided into IID dataset in 100 equal portions and allocate them to each client (IID dataset B in Table 2). Figure 9b plots the accuracy in rounds to exceed 80% as the amount of participating clients varies. In the experiment, there are only 2% difference when 100 clients participated in training.

### 4.4. Accuracy under Non-IID Traffic Distribution

We evaluate the accuracy of the FLIC under non-IID traffic distribution. As each client has different traffic data, 1 to 8 class data are distributed to 16 clients (Non-IID dataset A in Table 2). For example, in a three-class experiment, each client has different three traffic class data. Figure 10 shows the accuracy of FLIC under 1 to 8 class non-IID data for 16 clients. The accuracy of FLIC exceeds 80% after four class labels. From the experiment, we prove that FLIC achieves the high accuracy of Internet traffic classification even though the traffic data is not evenly distributed.

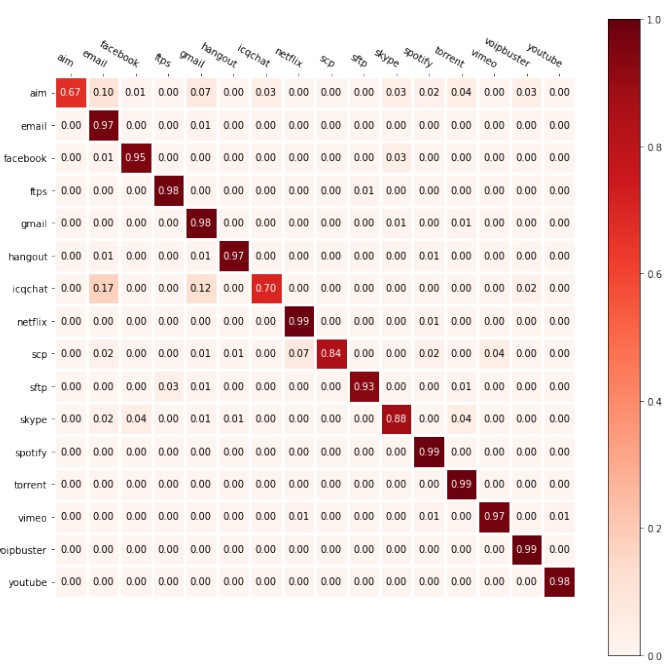

(**a**) Centralized learning

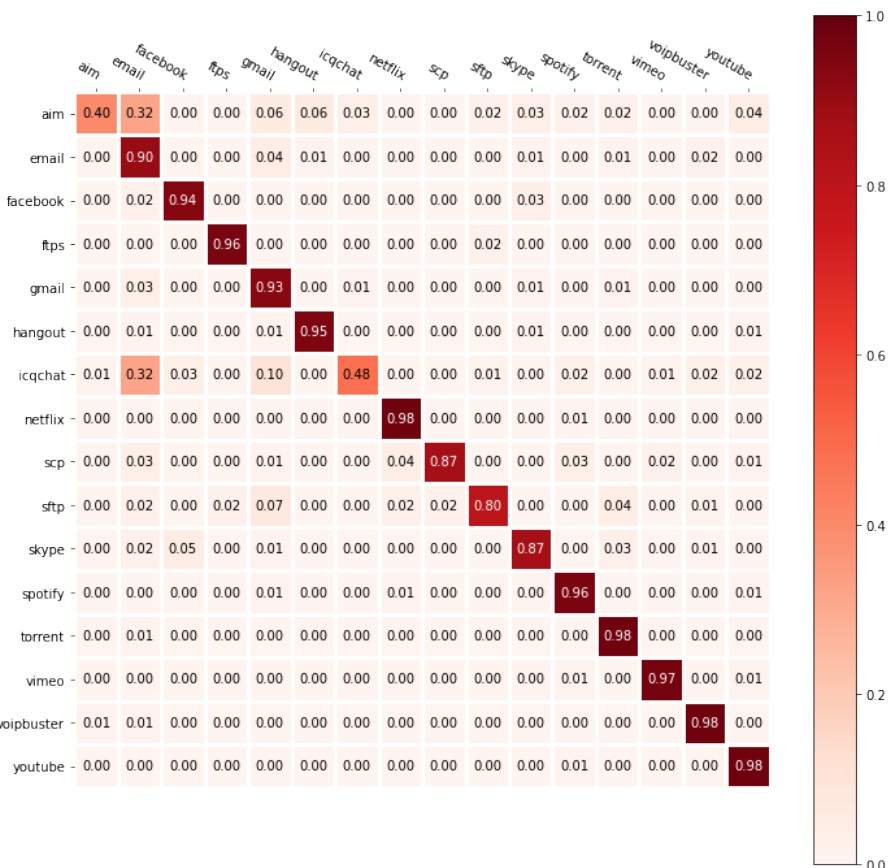

(**b**) Federated learning (Clients 32)

**Figure 8.** A confusion matrix of centralized learning and federated learning.

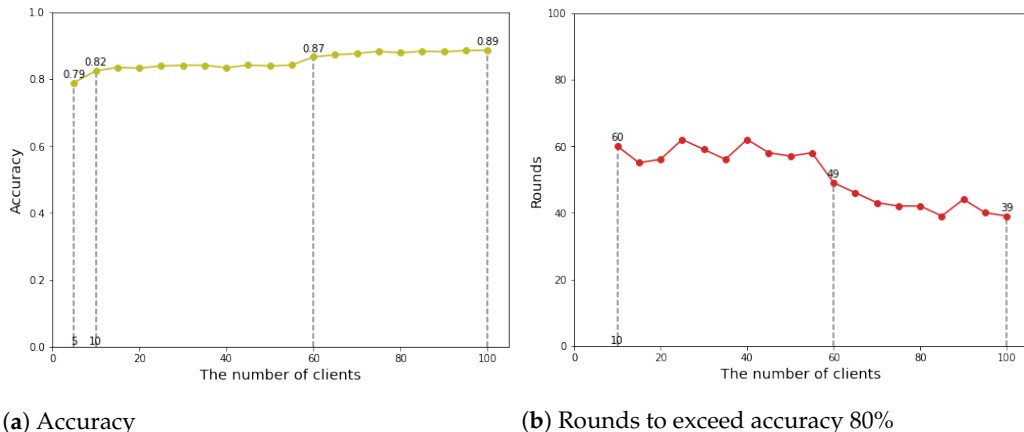

(**a**) Accuracy          (**b**) Rounds to exceed accuracy 80%

**Figure 9.** Accuracy of FLIC under the different number of clients participating in training.

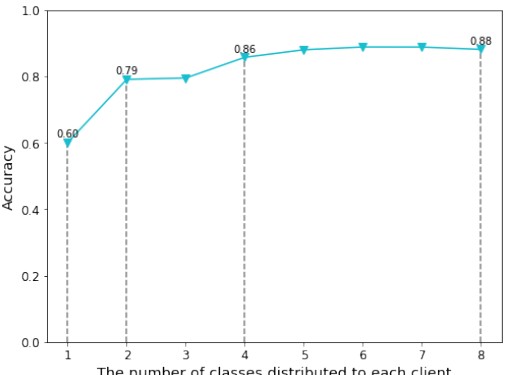

**Figure 10.** Accuracy of FLIC with the different number of classes distributed to one client (16 clients in total).

In federated learning, the data collected by the client is not transmitted to the central server, and only model parameters learned with their own data are sent to the central server. The data distribution for each client has a significant impact on the learning accuracy. Traffic data of popular services such as YouTube, Netflix, and Facebook are likely to be collected by many clients. However, each client has its own preferred application. Hence, the Internet traffic class labels of each client tend to be non-IID data.

As shown in Figure 11, when the non-IID feature becomes dominant, FLIC does not classify applications with little data. Looking at Figure 11a, which is a two-class non-IID confusion matrix, the global model of the two-class experiment categorizes YouTube and Netflix with a large amount of data well. However, FLIC does not classify `Aim` and `Icqchat` with a small amount of data. Even if the total amount of data distributed to all clients is the same, in the four-class non-IID experiment distributed to more clients (Figure 11b), FLIC classifies applications with a small amount of data better. This result is caused by the fact that the weights involved in classifying applications with a small amount of data are reduced to $1/n$ due to the federated average algorithm.

*4.5. Accuracy by Clients Local Epochs*

Local epochs indicate how many times a client learns the collected data when training the model with its own data and transmits the learned weights to the federated server. In centralized learning, there are no optimal solution under the low epochs, and over-fitting weights are computed under excessively high epochs. In federated learning, more rounds need to be performed to complete learning with lower epochs, and it will take longer learning time with the higher epochs. A round means that all clients train the model with their own data, and the training results are federated averaged on the federated server, so the higher the rounds, the longer the training time becomes.

We measure rounds with an accuracy of over 95% to see how many local epochs affect the learning rate in FLIC. We perform the experiment by changing local epoch from 1 to 400 using IID dataset A, which distributes sampled data to 32 clients.

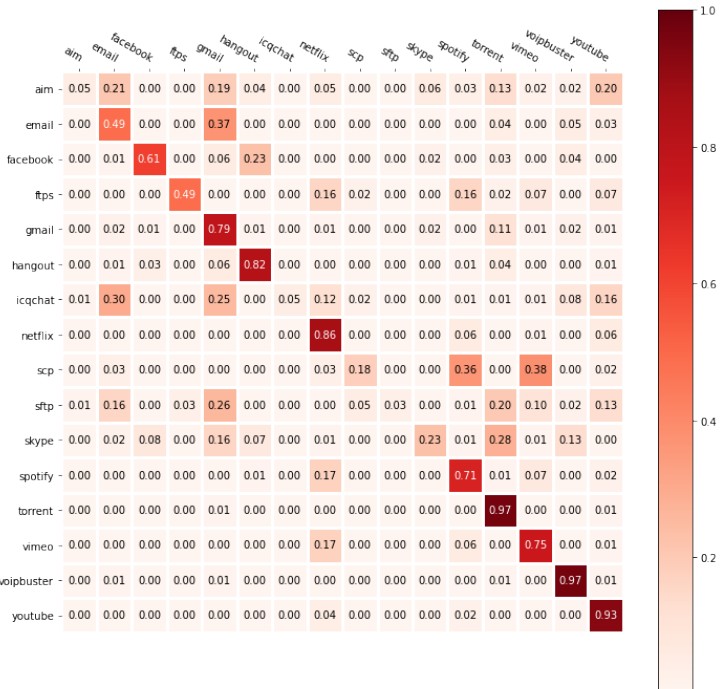

(**a**) 2-class non-IID data

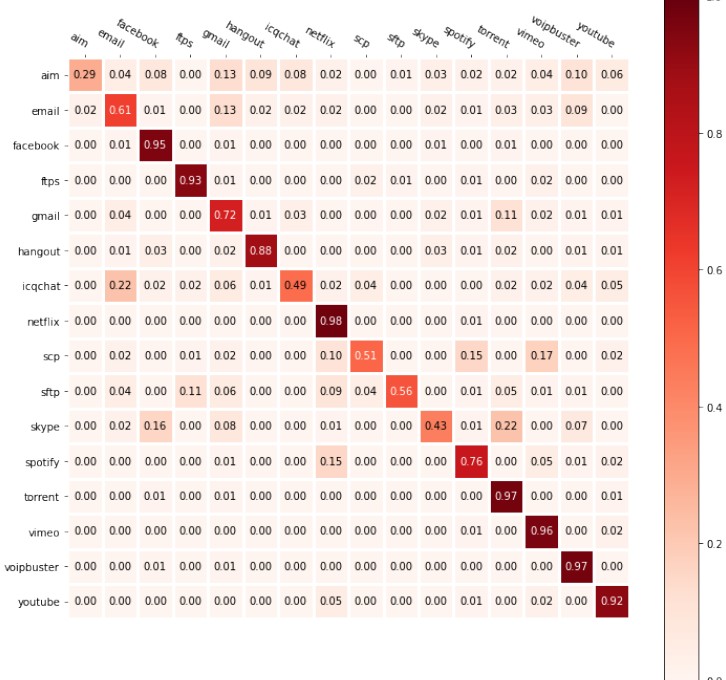

(**b**) 4-class non-IID data

**Figure 11.** A confusion matrix of federated learning under 2-class and 4-class non-IID data.

As shown in Table 3, as local epochs increase, the accuracy of 95% can be achieved in fewer rounds. In particular, as the local epochs increase, the rounds decrease, so when the local epoch is 30, 10.67 times fewer rounds can be completed.

**Table 3.** Rounds starting to exceed 95% accuracy by local epochs (16 clients).

| Local Epochs | Rounds | Local Epochs | Rounds |
|---|---|---|---|
| 1 | 64 (×1) | 30 | 6 (×10.67) |
| 2 | 33 (×1.94) | 40 | 8 (×8) |
| 3 | 19 (×3.37) | 50 | 8 (×8) |
| 4 | 18 (×3.56) | 60 | 6 (×10.67) |
| 5 | 16 (×4) | 70 | 6 (×10.67) |
| 6 | 14 (×4.57) | 80 | 6 (×10.67) |
| 7 | 11 (×5.82) | 90 | 5 (×12.8) |
| 8 | 14 (×4.57) | 100 | 5 (×12.8) |
| 9 | 12 (×5.33) | 200 | 5 (×12.8) |
| 10 | 10 (×6.4) | 300 | 5 (×12.8) |
| 20 | 7 (×9.14) | 400 | 5 (×12.8) |

*4.6. Dynamic Traffic Classification with Federated Learning*

In the real world, clients frequently join in the learning system or leave according to their environments. As each client has a different application usage pattern, federated learning should deal with classifying new Internet applications. FLIC using CNN exchanges classification labels for new applications and the density layer parameter for classification of the new application labels.

Compared with the previous studies, FLIC can support dynamic Internet traffic classification by adapting its training model and labels. We carry out an experiment by introducing new applications while new federated clients join in training. As shown in Figure 12, the experiment starts with two clients, and adds two new clients with two application labels every 10 rounds (non-IID dataset B in Table 2). When a new application label is included, the accuracy decreases for a while due to the initialization of the density layer parameter and the lack of information of the new class label. However, after FLIC aggregates the convolution layer parameter for the new application, its accuracy is gradually restored within two rounds. From this result, we prove that FLIC can achieve the accuracy of classifying new applications.

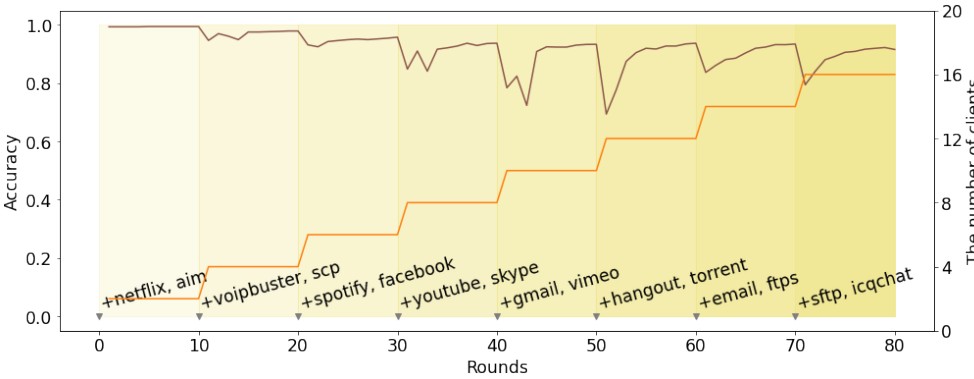

**Figure 12.** Accuracy of FLIC with dynamic clients and classification labeling classes.

Because we distribute each application data equally to two clients and participated in the learning, the result is similar to the two-class experiment in Section 4.4. However, unlike all clients from the beginning, in the dynamic experiment, two clients are added every 10 rounds. As a result, the accuracy of applications with less data, such as `AIM` and `ICQChat`, is lowered, but the accuracy of other applications is significantly improved compared to the two-class experiment (Figure 13). Under new applications, FLIC not only increases the number of applications that can be classified, but also proves better learning capability in the extreme non-IID data environment.

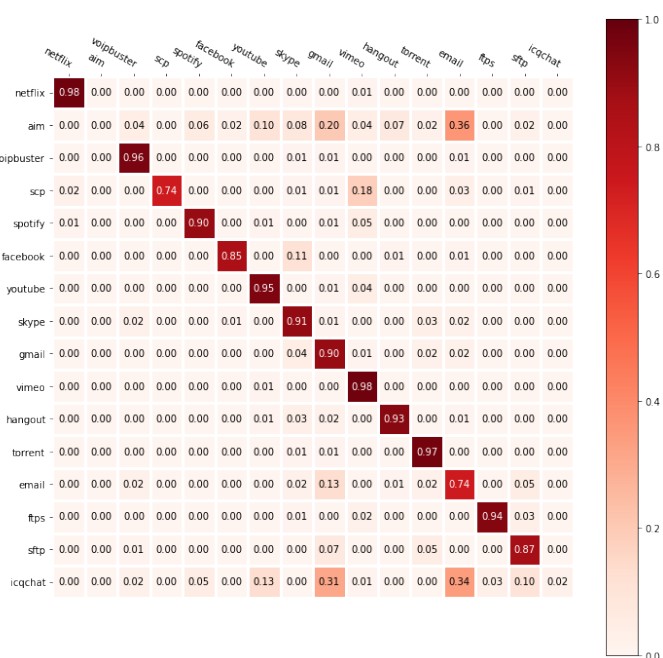

**Figure 13.** Confusion matrix of dynamic application federated learning.

## 5. Conclusions

In this paper, we propose a federated-learning Internet traffic classification framework (FLIC) that can classify packets into applications dynamically. In FLIC, the federated client directly trains its own packet data on the device, and sends only the result to the federated learning server. Since clients do not transmit their data to the server, personal information is protected, and as more clients participate, not only accuracy but also the number of Internet traffic applications that can be classified can be dynamically increased. Through comprehensive experiments with the TensorFlow federated library, we demonstrate that the more clients participate in training, the more accurate models can be trained. We have shown that FLIC can perform model training in a non-IID environment as well as dynamic Internet traffic application distribution. In the environment of non-IID traffic distribution and of dynamically increasing clients, FLIC achieved 88% and 92% accuracy, respectively. We solved the problem of a fault-tolerant client-server communication protocol federated learning framework that operates in an environment where the application to be classified is dynamically added through FLIC. We plan to study techniques such as FLIC behavior experiments in the real world, extreme non-IID (1-Class Non-IID) dataset learning methods, and differential privacy for additional personal information security reinforcement.

**Author Contributions:** Project administration, Y.L.; Software, H.M.; Writing—original draft, H.M.; Writing—review and editing, Y.L. All authors have read and agreed to the published version of the manuscript.

**Funding:** This work was supported by Institute of Information & communications Technology Planning & Evaluation(IITP) grant funded by the Korea government(MSIT) (No.2020-0-00901, Information tracking technology related with cyber crime activity including illegal virtual asset transactions)

**Conflicts of Interest:** The authors declare no conflict of interest.

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
