# Peer review of "Internet Traffic Classification with Federated Learning"

_electronics, doi:10.3390/electronics10010027_

Round 1

Reviewer 1 Report

The reviewed paper proposes a federated-learning traffic classification protocol FLIC, which can achieve the accuracy comparable to centralized deep learning for Internet application identification without privacy leakage. It is important to note that it can classify new applications on-the-fly when a participant joins in learning with a new application. Authors demonstrate that federated learning-based packet classification achieves high accuracy under non-independent and identically distributed traffic across clients.

The article is technically sound and the methodology used is acceptable. Machine learning constitutes the most important methodology component of the suggested solution. While Section 2 needs some elaboration and should discuss recent related works, their advantages and disadavantages, etc. Section 1 should provide wider background introduction to notions used in the paper, such as machine learning, deep learning and federated learning. In particular (line 29), introduction to a broad spectrum of machine learning uses and references are missing (incl. doi.org/10.3390/electronics9050821, doi.org/10.3390/electronics9020266, doi.org/10.3390/electronics9040689 and others).

In line 34, introduction to wide uses of federated learning with references is also missing (incl. doi.org/10.3390/electronics9050773, doi.org/10.3390/electronics9030440 etc.).

Another critical remark applies to inconsistency between general accuracy parameters in lines 17, 66 and 374, which needs to be made clear in the paper.

What is more, conclusions in Section 5 are too generic. A more in-depth discussion on what was achieved linking to the original research question should be provided. I would also like to see future study directions that authors want to assume basing on the results achieved so far.

The reference list seems to be too modest taking into account wide background and methodologies of the paper.

The paper needs language editing, in particular in case of articles. Very often definite article is used in wrong places (e.g. line 60). There are also missing spaces before opening brackets (e.g line 101 and 104).

I would also suggest that Figure 2 a/b is split in two separate figures to ensure proper graphics size and thus better readability. The same applies to Figures 8 a/b and 11 a/b. For example Figure 13 is much more readable.

All in all, the paper is interesting and the results shown are promising, which is why I would recommend that he paper be corrected as advised. I agree to re-review the paper to ensure the changes have been made.

Author Response

Dear MDPI Reviewers,

Thank you for reviewing our paper:

Paper title: “Internet Traffic Classification with Federated Learning”

First of all, we would like to appreciate your valuable comments. We devote ourselves to revise our manuscript based on all comments and reviews. This revision statement summarizes the detailed explanation of modifications. Please read the following.

We follow your comments and update the manuscript according to the reviews. In summary, the revisions that we have incorporated are as follows.

[Response to the Reviewer 1]

While Section 2 needs some elaboration and should discuss recent related works, their advantages and disadvantages, etc.

The reference list seems to be too modest taking into account wide background and methodologies of the paper.

  • Thank you for pointing out that although our study was influenced by many researchers, Secsion 2 related work was very poor. We divided the study into 4 main parts and reinforced the related research; Traffic classification using machine learning, federated learning, client selection in federated learning, federated learning in a non-IID environment. Detailed changes are as follows. The line number is a number in the revised draft.
    • Secsion 2 overall reorganization. (Blue emphasis)
    • Paragraph 1, lines 71-77. Traffic classification research status
    • Paragraph 2, lines 78-87. Traffic classification study using CNN.
    • Paragraph 3, lines 88-92. Comparison of traffic classification studies using CNN / RNN.
    • Paragraph 4, lines 93-101. Federated learning.
    • Paragraph 5, sentences 2, 3, and 4. lines 104-108. Federated learning algorithm.
    • Paragraph 6, sentences 3 and 4. lines 112-117. Client selection in Federated learning
    • Paragraph 7, sentences 4 and 5. lines 123-126. Non-IID environments
    • Paragraph 8. lines 129-136. Traffic classification study using Federated learning
    • Paragraph 9. lines 137-140. Deep learning model layer sharing.

Section 1 should provide wider background introduction to notions used in the paper, such as machine learning, deep learning and federated learning.
In particular (line 29), introduction to a broad spectrum of machine learning uses and references are missing (incl. doi.org/10.3390/electronics9050821, doi.org/10.3390/electronics9020266, doi.org/10.3390/electronics9040689 and others).

In line 34, introduction to wide uses of federated learning with references is also missing (incl. doi.org/10.3390/electronics9050773, doi.org/10.3390/electronics9030440 etc.).

  • Thank you for pointing out in the Secsion 1 outline that we did not provide an overview of the technology base on which our research is located. We have more clearly indicated the background along with the 5 papers provided.
    • Line 30~31. Indicate that machine learning is being used in a variety of applications.
    • Line 39. Indicates that federated learning is being used in a distributed environment.

Another critical remark applies to inconsistency between general accuracy parameters in lines 17, 66 and 374, which needs to be made clear in the paper.

  • Thank you for pointing out that we didn't write the numbers accurately. Although numbers are very important in scientific papers, we recognized that numbers were poorly matched during paper writing and reviewed all of the overall numbers in the paper.
    • Line 16~19, line 68~69, line 429~430. Check numbers in abstract, main text, and conclusion.

What is more, conclusions in Section 5 are too generic. A more in-depth discussion on what was achieved linking to the original research question should be provided. I would also like to see future study directions that authors want to assume basing on the results achieved so far.

  • Thank you for pointing out that our conclusions only present what we have done, what problems we solved and what we are going to do in the future. We clearly expressed which problems FLIC solved, and also presented problems to be solved in the future.
    • Line 430~435. Expressing the problem solved and the problem to be solved

The paper needs language editing, in particular in case of articles. Very often definite article is used in wrong places (e.g. line 60). There are also missing spaces before opening brackets (e.g line 101 and 104).

  • Thank you for mentioning your carelessness regarding editing. We looked again and corrected things like parentheses.
  • In addition, the papers that used the ISCXVPN2016 data set for the first time were clearly expressed, and the sentences were corrected to avoid misunderstandings.
    • Line 63~64. ISXCVPN2016 paper introduction.
    • Line 161, 174, 183, 204, 205, etc. Double check space before parenthesis

I would also suggest that Figure 2 a/b is split in two separate figures to ensure proper graphics size and thus better readability. The same applies to Figures 8 a/b and 11 a/b. For example Figure 13 is much more readable.

  • Thank you for giving us good advice on the readability of the pictures. We knew that the readability of Figures 2, 8, and 11 was low, but while we were thinking about which method to use, the readability of the figure improved further with good advice. As in Fig. 13, Figs. 2, 8, and 11 are all enlarged to 1 column.
    • Page 7, figure 2. Magnified by Figure 13.
    • Page 13, figure 8. Magnified by Figure 13.
    • Page 15, figure 11. Magnified by Figure 13.
    • Page 17, figure 13. Match horizontal size with Figures 2, 8, and 11

Reviewer 2 Report

  1. This paper proposes a federated-learning traffic classification protocol (FLIC) for the Internet application identification without privacy leakage.
  2. More references are necessary to support your key points.
  3. To distance you from your work and maintain objectivity, it is generally best to avoid using the first person (beginning sentences with "We") in your paper. As a general rule, limit your use of the first person to when you want to express your opinion, such as “we propose,” “we check,” “we present,” “we explore,” “we simulate,” “we attempt, ”“we challenge, ” “we investigate, ” “we acknowledge, ” “we estimate, ” “we select ,” “we decide,” “we reduce,” “we set,” “we use,” “we know,” “we find,” “we search,” "we believe," "we infer," "we conclude," "we postulate," "we hypothesize," “we base,” “we seek,” or “we organize.”

Author Response

Dear MDPI Reviewers,

Thank you for reviewing our paper:

Paper title: “Internet Traffic Classification with Federated Learning”

First of all, we would like to appreciate your valuable comments. We devote ourselves to revise our manuscript based on all comments and reviews. This revision statement summarizes the detailed explanation of modifications. Please read the following.

We follow your comments and update the manuscript according to the reviews. In summary, the revisions that we have incorporated are as follows.

[Response to the Reviewer 2]

More references are necessary to support your key points.

  • Thank you for pointing out that although our study was influenced by many researchers, Secsion 2 related work was very poor. We divided the study into 4 main parts and reinforced the related research; Traffic classification using machine learning, federated learning, client selection in federated learning, federated learning in a non-IID environment. Detailed changes are as follows. The line number is a number in the revised draft.
    • Secsion 2 overall reorganization. (Blue emphasis)
    • Paragraph 1, lines 71-77. Traffic classification research status
    • Paragraph 2, lines 78-87. Traffic classification study using CNN.
    • Paragraph 3, lines 88-92. Comparison of traffic classification studies using CNN / RNN.
    • Paragraph 4, lines 93-101. Federated learning.
    • Paragraph 5, sentences 2, 3, and 4. lines 104-108. Federated learning algorithm.
    • Paragraph 6, sentences 3 and 4. lines 112-117. Client selection in Federated learning
    • Paragraph 7, sentences 4 and 5. lines 123-126. Non-IID environments
    • Paragraph 8. lines 129-136. Traffic classification study using Federated learning
    • Paragraph 9. lines 137-140. Deep learning model layer sharing.

To distance you from your work and maintain objectivity, it is generally best to avoid using the first person (beginning sentences with "We") in your paper. As a general rule, limit your use of the first person to when you want to express your opinion, such as “we propose,” “we check,” “we present,” “we explore,” “we simulate,” “we attempt, ”“we challenge, ” “we investigate, ” “we acknowledge, ” “we estimate, ” “we select ,” “we decide,” “we reduce,” “we set,” “we use,” “we know,” “we find,” “we search,” "we believe," "we infer," "we conclude," "we postulate," "we hypothesize," “we base,” “we seek,” or “we organize.”

  • Thank you for pointing out our English grammar. We are aware that our presentation is not professional and plan to apply for the MDPI english grammar check service. However, this review needs to be processed in 4 days, so we used the English grammar correction service that we can use quickly. In particular, I know that the sentences from the first person perspective need to be intensively corrected, but the results have not yet arrived, so only a few corrections were applied. That revision was not highlighted in blue.
    • Line 16. Also -> In addition
    • Line 28. Instead of it -> Insteads
    • Line 51. make -> devise
    • Line 55. user -> client
    • Line 59. transits -> sends
    • Line 118. As well as -> In addition to
    • Line 112. reduced by -> decreased
    • Line 126. According to the loss, -> when [7]
    • Line 151. collects data for training -> collects data
    • Line 155. training result -> training results
    • Line 155. aggregates it -> aggregates them
    • Line 179. client selection and waiting -> client selection
    • Line 180. As the FLIC server cannnot -> As the FLIC server does not
    • Line 187. federated client can -> federated client are able to
    • Line 190. tshark and then it vectorizes -> tshark, and it vectorizes
    • Line 209 and 911. numbers -> values
    • Line 247. first 1, 2, 4 -> first 1, 2, and 4
    • Line 253. show visual pattern -> show the visual pattern
    • Line 257. Federated learning works -> Federated learning performs
    • Line 283. FLIC can dynamically increase application classification targets -> FLIC dynamically adds new application targets
    • Line 296, 305. proprocess -> preprocess
    • Line 297. is saved in an image format such as jpg or png -> is saved in jpg or png
    • Line 298. so it must be saved in 2D-array -> We save the image in 2D-array
    • Line 298. we partitioned -> we partition
    • Line 299. two sets of IID dataset and two sets of non-IID dataset -> two IID datasets and two non-IID datasets
    • Line 302. When creating a model, the generator create -> The generator creates
    • Line 306. The simulator implemented -> We implemented the simulator
    • Line 323. each dataset -> each data

Reviewer 3 Report

This is a great paper. Strong accept.

One thing can be further improved is the 'related' work. It would be beneficial to introduce federated learning in related work.

Author Response

[Response to the Reviewer 3]

One thing can be further improved is the 'related' work. It would be beneficial to introduce federated learning in related work.

  • Thank you for pointing out that although our study was influenced by many researchers, Secsion 2 related work was very poor. We divided the study into 4 main parts and reinforced the related research; Traffic classification using machine learning, federated learning, client selection in federated learning, federated learning in a non-IID environment. Detailed changes are as follows. The line number is a number in the revised draft.
    • Secsion 2 overall reorganization. (Blue emphasis)
    • Paragraph 1, lines 71-77. Traffic classification research status
    • Paragraph 2, lines 78-87. Traffic classification study using CNN.
    • Paragraph 3, lines 88-92. Comparison of traffic classification studies using CNN / RNN.
    • Paragraph 4, lines 93-101. Federated learning.
    • Paragraph 5, sentences 2, 3, and 4. lines 104-108. Federated learning algorithm.
    • Paragraph 6, sentences 3 and 4. lines 112-117. Client selection in Federated learning
    • Paragraph 7, sentences 4 and 5. lines 123-126. Non-IID environments
    • Paragraph 8. lines 129-136. Traffic classification study using Federated learning
    • Paragraph 9. lines 137-140. Deep learning model layer sharing.

Reviewer 4 Report

Overall the topic is timely. I have the following main concerns:

  • Eq.(1), and (2) require appropriate references. 
  • The Federated optimization model is missing a regularizer function. 
  • A summary of mathematical notations can further help. Careful proofreading is needed.
  • The proposed framework seems to be beneficial in all figures, there is not a single trade-off? This is strange. It is suggested to clearly indicate the trade-off associated with the proposed framework.

Author Response

Dear MDPI Reviewers,

Thank you for reviewing our paper:

Paper title: “Internet Traffic Classification with Federated Learning”

First of all, we would like to appreciate your valuable comments. We devote ourselves to revise our manuscript based on all comments and reviews. This revision statement summarizes the detailed explanation of modifications. Please read the following.

We follow your comments and update the manuscript according to the reviews. In summary, the revisions that we have incorporated are as follows.

[Response to the Reviewer 4]

Eq.(1), and (2) require appropriate references. The Federated optimization model is missing a regularizer function. A summary of mathematical notations can further help. Careful proofreading is needed.

  • Thank you for pointing out the lack of references to our equation. We have clearly marked the paper published by Google, clarifying the source of the formula. In addition, the related research on the federated learning algorithm was described by reinforcing the related work as detailed below. The line number is a number in the revised draft.
    • Secsion 2 overall reorganization. (Blue emphasis)
    • Paragraph 1, lines 71-77. Traffic classification research status
    • Paragraph 2, lines 78-87. Traffic classification study using CNN.
    • Paragraph 3, lines 88-92. Comparison of traffic classification studies using CNN / RNN.
    • Paragraph 4, lines 93-101. Federated learning.
    • Paragraph 5, sentences 2, 3, and 4. lines 104-108. Federated learning algorithm.
    • Paragraph 6, sentences 3 and 4. lines 112-117. Client selection in Federated learning
    • Paragraph 7, sentences 4 and 5. lines 123-126. Non-IID environments
    • Paragraph 8. lines 129-136. Traffic classification study using Federated learning
    • Paragraph 9. lines 137-140. Deep learning model layer sharing.

The proposed framework seems to be beneficial in all figures, there is not a single trade-off? This is strange. It is suggested to clearly indicate the trade-off associated with the proposed framework.

  • Thank you for pointing out that the stateless FLIC does not specify even though it has obvious disadvantages. We understood that almost all studies have shortcomings, but they are not specified, and are expressed as though there are no problems to be resolved later. Therefore, we have expressed our shortcomings by modifying as follows, and also explained the plans to be resolved in the future.
    • Line 193~197. Add disadvantages of FLIC
    • Line 430~435. Expressing the problem solved and the problem to be solved
